# On Learning Multi-Modal Forgery Representation for Diffusion Generated Video Detection

**Xiufeng Song**[1], **Xiao Guo**[2], **Jiache Zhang**[1], **Qirui Li**[1],
**Lei Bai**[3], **Xiaoming Liu**[2], **Guangtao Zhai**[1], **Xiaohong Liu**[*1]

[1]Shanghai Jiao Tong University [2]Michigan State University [3]Shanghai Artificial Intelligence Laboratory
{akikaze, zjc_he, iapple1, zhaiguangtao, xiaohongliu}@sjtu.edu.cn
{guoxia11, liuxm}@cse.msu.edu baisanshi@gmail.com
[*] Corresponding Author

## Abstract

Large numbers of synthesized videos from diffusion models pose threats to information security and authenticity, leading to an increasing demand for generated content detection. However, existing video-level detection algorithms primarily focus on detecting facial forgeries and often fail to identify diffusion-generated content with a diverse range of semantics. To advance the field of video forensics, we propose an innovative algorithm named Multi-Modal Detection(MM-Det) for detecting diffusion-generated videos. MM-Det utilizes the profound perceptual and comprehensive abilities of Large Multi-modal Models (LMMs) by generating a Multi-Modal Forgery Representation (MMFR) from LMM's multi-modal space, enhancing its ability to detect unseen forgery content. Besides, MM-Det leverages an In-and-Across Frame Attention (IAFA) mechanism for feature augmentation in the spatio-temporal domain. A dynamic fusion strategy helps refine forgery representations for the fusion. Moreover, we construct a comprehensive diffusion video dataset, called Diffusion Video Forensics (DVF), across a wide range of forgery videos. MM-Det achieves state-of-the-art performance in DVF, demonstrating the effectiveness of our algorithm. Both source code and DVF are available at link.

## 1 Introduction

Recent years have witnessed significant advancements in diffusion generative methods, which have led to the creation of extraordinarily visually compelling content in video generation [5, 4, 61]. Although the latest generated videos impress society with their versatility and stability, synthetic media also poses a risk of malicious attacks, such as counterfeit faces created by deepfakes [49] and falsifications in business, raising public concerns about information security and privacy. In response to such issues, researchers have made significant progress in forgery detection, addressing problems on image editing manipulation [60, 30, 17] and CNN-synthesized images [53, 40, 55, 36, 18]. To enhance the trustworthiness and reliability of current detectors in the face of evolving generative video methods, we aim to develop a generalizable detection method for diffusion-based generative videos.

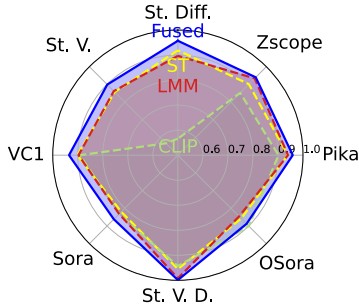

Figure 1: Multi-Modal Detection (MM-Det) leverages features from spatiotemporal (ST) information (🟨), a CLIP encoder [39] (🟩), and an LMM (🟥). The Fused feature (🟦) achieves state-of-the-art performance in our Diffusion Video Forensics (DVF) dataset.

38th Conference on Neural Information Processing Systems (NeurIPS 2024).

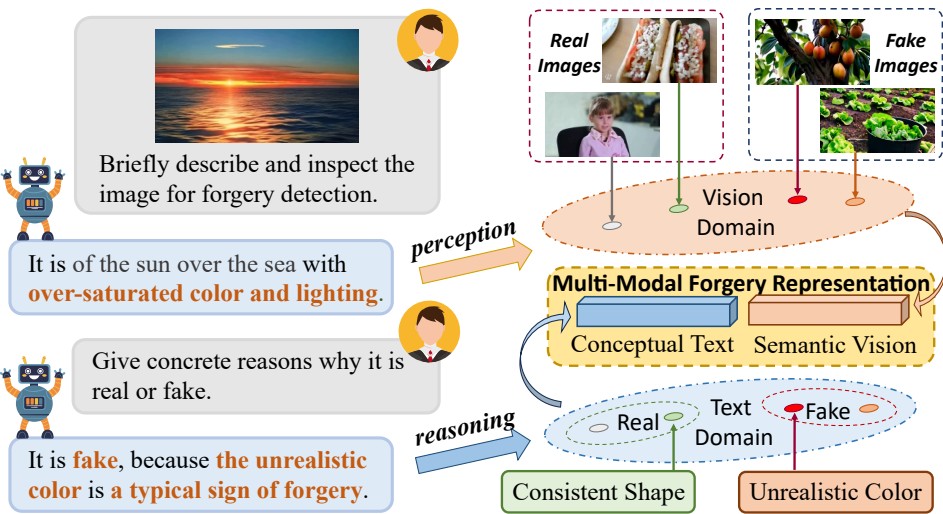

Figure 2: LMMs detect visual artifacts and anomalies, offering detailed textual reasoning that explains whether the image is generated using Artificial Intelligence (AI) techniques. The powerful representation in the visual domain enables LMMs to understand complex contexts within frames. Furthermore, their advanced language reasoning capability implicitly reveals image authenticity and provenance. For instance, the term like ``consistent shape'' refers to common features in authentic content, while ``unrealistic color'' signifies typical artifacts in forged content. This linguistic proficiency stems from the superior perception and comprehension abilities of LMMs, contributing to a generalizable multimodal feature space. By leveraging the visual understanding and textual reasoning abilities of LMMs, we construct a Multi-Modal Forgery Representation (MMFR).

Previously, the video forensics community emphasized more on developing facial forgery detection algorithms [57, 73, 10, 62, 70], which may struggle to address recent fraudulent videos (*e.g.*, sora, pika, etc.). Compared to facial forgery, diffusion-based generated content contains more diverse semantics, making it more challenging to distinguish diffusion forgery contents from real ones. Towards these challenges, a new thread of research on CNN-generated image detection has emerged [53, 40, 55, 36, 18]. These works aim to learn common generation traces in image-level content, but do not design specific mechanisms to capture temporal inconsistencies in videos.

Therefore, previous defensive efforts might not be able to provide a video-level detection algorithm for newly emerged generated videos with diverse manipulation artifacts and visual contexts. Meanwhile, Large Multi-modal Models (LMMs) show unparalleled problem-solving ability [2, 24, 23, 75, 69, 28], thanks to its powerful multi-modal representations. However, such representations are barely studied in the video forensics task.

Motivated by the limitation of previous work and the unprecedented understanding ability of LMMs, we propose a video-level detection algorithm, named Multi-Modal Detection (MM-Det), to capture forgery traces based on an LMM-based multi-modal representation. MM-Det takes advantage of the perception and reasoning capabilities of LMMs to learn a generalizable forgery feature, as depicted in Fig. 2. To the best of our knowledge, we are the first to use LMMs for video forensic work.

Aside from multi-modal representations, two common sources of generative errors that can be leveraged for video discrimination are spatial artifacts and temporal inconsistencies. Our approach aims to effectively identify these two types of errors as an auxiliary feature in forgery detection. Inspired by the previous work [55, 33, 41] that shows the effectiveness of reconstruction for detecting diffusion images, we extend this idea into the video domain, amplifying diffusion artifacts both in spatial and temporal information. To capture such artifacts efficiently, we leverage a Vector Quantised-Variational AutoEncoder (VQ-VAE) [51] for a fast reconstruction process, as detailed in Fig. 3. Moreover, we design a novel **I**n-and-**A**cross **F**rame **A**ttention (IAFA) into a Transformer-based network, which balances frame-level forgery traces with information flow across frames, thus aggregating local and global features.

Although diffusion methods demonstrate strong capabilities in video generation, the lack of public datasets on diffusion videos hinders research efforts in the video forensic community. In light of this,

we have established a comprehensive dataset for diffusion-generated videos, named Diffusion Video Forensics (DVF). DVF includes generated content from a variety of diffusion models, featuring rich semantics and high quality, serving as a general benchmark for open-world video forensics tasks. The main contributions of this paper are as follows:

◇ We propose a detection method called MM-Det that leverages a **M**ulti-**M**odal Forgery Representation from LMMs to effectively detect diffusion-generated videos with strong generalization capability.

◇ A powerful and innovative In-and-Across Frame Attention (IAFA) mechanism is introduced to aggregate global and local patterns within forged videos, enhancing the detection of spatial artifacts and temporal inconsistencies.

◇ We introduce a large-scale dataset, named the Diffusion Video Forensics (DVF) dataset, comprising high-quality forged videos generated using 8 diffusion-based methods. The DVF dataset contains diverse forgery types across videos of varying resolutions and durations, effectively serving as a benchmark for forgery detection in real-world scenarios.

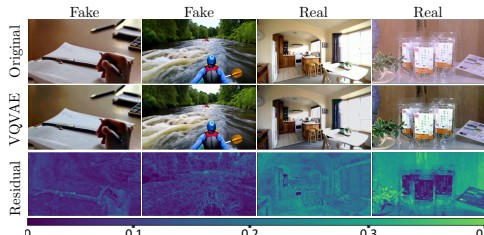

Figure 3: The residual difference between VQ-VAE [51] reconstructed images and real ones. Given an encoder $\mathcal{E}$ and a decoder $\mathcal{D}$ of a VQ-VAE and taking the input video $\mathbf{v}$, the reconstructed video $\mathbf{v}'$ is obtained as $\mathbf{v}' = \mathcal{D}(\mathcal{E}(\mathbf{v}))$. The VQ-VAE reconstruction of real images exhibits obvious edges and visible traces, whereas diffusion-generated ones are reconstructed more effectively, offering residual difference images with fewer visible traces.

◇ Our MM-Det achieves state-of-the-art detection performance on the DVF dataset. Also, a detailed analysis is provided to showcase the effectiveness of multi-modal representations in detecting forgeries, paving the way for compelling opportunities for using LMMs in future multi-media forensic research.

## 2 Related Works

**Frame-level Detector** Early work [53, 22, 13, 47, 65, 16] observed that forgery traces exist in images generated by AI techniques, and such traces are commonly used as evidence to distinguish diffusion-generated content [40, 6, 7] and attribute if two images are generated by the same method [66, 37]. However, identifying unseen and diverse frequency-based clues in real-world scenarios is challenging. For that, existing frame-level forgery detectors concentrate on improving the generalization ability. For example, some works [36, 8, 17, 26, 29] introduced features from pre-trained CLIP [39] encoders for the forensic task to help reduce the overfitting issue on specific forgery types and increase the robustness towards detection [36, 8] and localization [17]. [64] and [12] proposed proactive methods to protect images from manipulation based on image watermarking and steganography. Also, reconstruction errors through the inversion process of DDIM [44] are studied by prior works [55, 33, 41, 32] for diffusion generative content detection. Moreover, the previous work [63, 46, 34, 18] develops specific techniques that increase the generalization to unseen forgeries. For example, HiFi-Net [18] proposes a tree structure to model the inherent hierarchical correlation among different forgery methods, NPR [46] devises a special representation as generative artifacts, and the training set diversity can also contribute to generalization ability [34]. Unlike prior works, our MM-Det leverages multi-modal reasoning to achieve a high level of generalization ability.

**Video-level Detector** Early video-level methods primarily focused on detecting facial forgery. For example, [25] learned the boundary artifacts between the original background and manipulated faces. [19] discriminated fake videos from the inconsistency of mouth motion. [73] designed a multi-attentional detector to capture deepfake features and artifacts. F3Net [38] captured global and local forgery traces in the frequency domain. [10, 15, 74, 56] explored temporal information and inconsistency from fake videos. Most recently, DD-VQA [70] formulates deepfake detection as a sentence-generation problem, largely improving the interpretation of deepfake detection. However, these studies are restricted to facial forgery methods, which are insufficient for the current defensive systems that address diverse content produced by diffusion models. Therefore, we develop MM-Det to detect diffusion video content, pushing forward the frontier of forgery video detection.

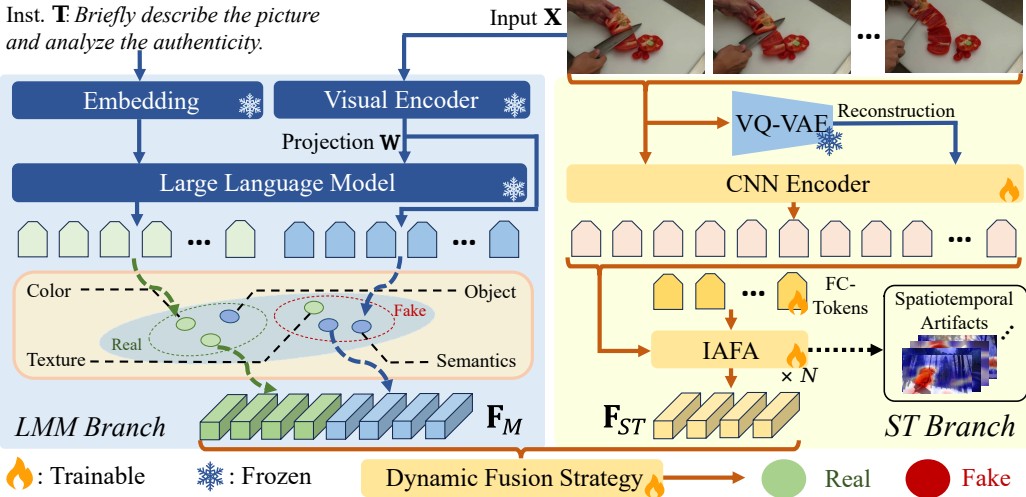

Figure 4: Multi-Modal Detection network (MM-Det) architecture. Given an input video $\mathbf{X}$, the Large Multi-modal Model (LMM) branch takes the frame and instructions to generate Multi-Modal Forgery Representation (MMFR). Hidden states from the visual encoder and large language model are extracted to form the MMFR, denoted as $\mathbf{F}_M$, which helps capture the forgery traces among different diffusion-generated videos. In the Spatio-Temporal (ST) branch, videos are first reconstructed via a VQ-VAE, and then fed into a CNN encoder, followed by In-and-Across Frame Attention (IAFA) modules detailed in Sec. 3.2. IAFA is introduced to capture features based on spatial artifacts and temporal inconsistencies, termed as $\mathbf{F}_{ST}$. At last, a dynamic fusion strategy combines $\mathbf{F}_M$ and $\mathbf{F}_{ST}$ for the final forgery prediction.

**Large Multi-modal Models (LMMs)** LMMs possess generalizable problem-solving abilities in real-world tasks, including object detection[14], semantic segmentation[67] and visual question answering[59]. [2, 23, 27, 28] studied feature alignment schemes to bridge visual and textual domains for LMMs. [68, 72, 71] extended the boundaries of LMMs to multi-modal downstream tasks. [45] aligns multi-modal features for capabilities on cross-domain behaviors. [58] developed a Large Language Model (LLM)-based feature extractor for cheap-fake detection. Inspired by these studies, we stimulate the powerful perceptual and reasoning ability of an LMM by introducing the multi-modal feature space in video forgery detection.

## 3 Methods

In this section, we introduce the Multi-Modal Detection (MM-Det) framework for diffusion video detection, as depicted in Fig. 4. More formally, Sec. 3.1 details a Large Multi-modal Model (LMM) branch that learns a Multi-Modal Forgery Representation (MMFR). Then Sec. 3.2 reports a Spatio-Temporal (ST) branch that utilizes In-and-Across Frame Attention (IAFA) to capture spatial artifacts and temporal inconsistencies in forged videos. Lastly, a dynamic fusion technique reported in Sec. 3.3 adaptively combines outputs from the LMM branch and ST branch.

### 3.1 Multi-Modal Forgery Representation

We propose a novel Multi-Modal Forgery Representation (MMFR) from the multi-modal space of LMMs in LMM branch. This representation utilizes the powerful perceptual and reasoning abilities of LMMs in the form of instruction-based conversations.

Specifically, LMM branch is built on the top of LLaVA [28], one representative LMM, which has two key components: a visual encoder (*e.g.*, $\mathcal{D}_v$), instantiated by visual encoders from the Contrastive Language-Image Pre-Training (CLIP) [39], and the large language model $\mathcal{D}_L$ (*i.e.*, Llama 2 [50]). Let us denote the input video as $\mathbf{X} \in \mathbb{R}^{N \times H \times W \times C}$ that contains $N$ frames, where each frame is represented as $\mathbf{x} \in \mathbb{R}^{H \times W \times C}$. First, $\mathbf{x}$ is fed to $\mathcal{D}_v$ to obtain the corresponding visual representation $\mathbf{F}_V \in \mathbb{R}^Z$. Such $\mathbf{F}_V$ not only contains rich semantics but also shows impressive generalization ability and robustness in the forgery detection task [36, 43, 9]. Then, a textual instruction $\mathbf{T}$ is sampled from

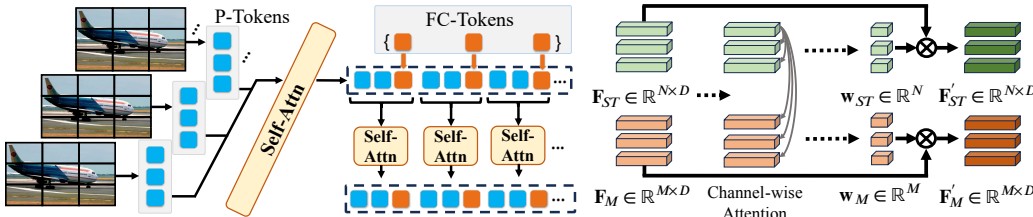

(a) Mechanism of In-and-Across Frame Attention      (b) Dynamic Fusion Strategy

Figure 5: (a) In-and-Across Frame Attention (IAFA): Each input frame (or its feature map) is divided into patches that are transformed into tokens, termed P-tokens (■). We introduce additional frame-centric tokens FC-tokens (■) encapsulating the global forgery information of the video frame. In each transformer layer, self-attention is applied alternately among all P-tokens from video frames as well as among the same frame's P-tokens and its FC-tokens. (b) The dynamic fusion strategy captures that takes $\mathbf{F}_{ST}$ and $\mathbf{F}_M$ as inputs and output channel-wise dependencies, which help refine forgery representations for the fusion.

pre-defined templates $\mathbf{Q}$ to guide the LMM on forgery detection reasoning. Both visual representation $\mathbf{F}_V$ and instruction $\mathbf{T}$ are fed to $\mathcal{D}_L$, which generates enhanced visual representations (*e.g.*, $\mathbf{F}_L$). We convert $\mathbf{F}_V$ into a sequence of visual tokens [28] (*i.e.*, $\mathbf{H}_v = \{\mathbf{h}_{v,m}\}_{m=1}^{M} \in \mathbb{R}^{M \times D}$), and $\mathbf{T}$ is transformed into textual tokens $\mathbf{H}_t = \{\mathbf{h}_{t,o}\}_{o=1}^{O} \in \mathbb{R}^{O \times D}$. Both $\mathbf{H}_v$ and $\mathbf{H}_t$ are taken as the input to $\mathcal{D}_L$, generating $\mathbf{F}_L \in \mathbb{R}^{S \times D}$ that can be tokenized into the language response providing reasoning (Fig. 2) about the authenticity of the input $\mathbf{x}$. This procedure is formulated as

$$\mathbf{F}_L = \mathcal{D}_L(\mathbf{H}_t, \mathbf{H}_v) = \mathcal{D}_L(\mathbf{T}, \mathbf{F}_V), \tag{1}$$

where $\mathbf{T}$ guides the pre-trained $\mathcal{D}_L$ in comprehending visual content (*i.e.*, $\mathbf{F}_V$), discerning the subset information from $\mathbf{F}_V$. This instruction $\mathbf{T}$ enables LMM branch to obtain the multi-modal representation that leverages the generalization ability from the pre-trained large language model Llama 2 (*i.e.*, $\mathbf{D}_L$), being different to prior work [36, 9] that only relies on $\mathbf{F}_E$.

Lastly, we retrieve the final MMFR, denoted as $\mathbf{F}_M \in \mathbb{R}^{M \times Z}$, by concatenating $\mathbf{F}_V$ and $\mathbf{F}_L$ after a linear layer(*i.e.*, PROJ), as

$$\mathbf{F}_M = \texttt{CONCAT}(\{\texttt{PROJ}(\mathbf{F}_V), \texttt{PROJ}(\mathbf{F}_L)\}). \tag{2}$$

### 3.2 Capturing Spatial-Temporal Forgery Traces

Targeting capturing spatiotemporal artifacts in video tasks, we introduce a Spatial-Temporal (ST) branch that learns effective diffusion forgery representation at the video level. Through a reconstruction procedure, we amplify the diffusion traces in the frequency domain, which is then captured by In-and-Across Frame Attention (IAFA) to form an effective video-level feature.

**Amplification of Diffusion Traces** Similar to prior studies [55, 33] that discovered specific generative traces of diffusion models through reconstruction on diffusion-generated images, we utilize an Autoencoder to amplify diffusion traces in videos. The reconstruction procedure can be expressed as follows.

More formally, denote the input video as $\mathbf{X} \in \mathbb{R}^{N \times H \times W \times C}$ that contains $N$ frames, in which each frame is represented as $\mathbf{x} \in \mathbb{R}^{H \times W \times C}$. We leverage a VQ-VAE [51] to obtain the reconstructed version of $\mathbf{x}$, which is denoted as $\hat{\mathbf{x}} \in \mathbb{R}^{H \times W \times C}$. The difference between $\mathbf{x}$ and $\hat{\mathbf{x}}$ makes an effective indicator of showing if the input is generated by diffusion models, as depicted in Fig. 3. Therefore, we jointly proceed $\mathbf{x}$ and $\hat{\mathbf{x}}$ to the following proposed modules for learning an effective representation of discerning forgeries.

It is worth mentioning that the prior approach [55] also adopts the idea of using the residual difference between original and reconstructed inputs to help forgery detection, but the reconstruction method requires multiple time-step denoising operations, which are computationally infeasible to reconstruct all frames from $\mathbf{X}$. In contrast, our VQ-VAE-based reconstruction method only requires one single forward propagation to obtain reconstructed frame $\hat{\mathbf{x}}$, meanwhile preserving the effectiveness in indicating the discrepancy between real and fake inputs.

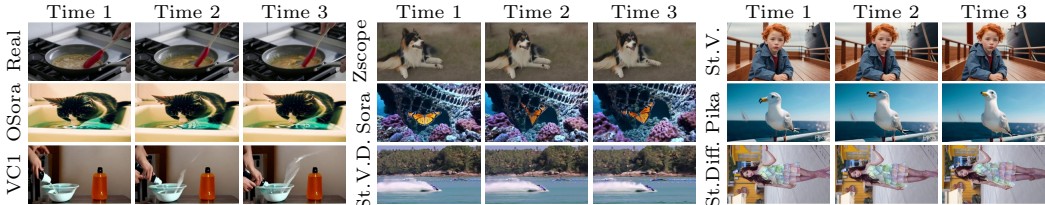

Figure 6: Sampled videos from DVF dataset. DVF contains 8 video generation methods, including 7 text-to-video methods and 1 image-to-video method. Real videos are selected from Internvid-10M [54] and Youtube-8M [1]. [Key: OSora: OpenSora; VC1: Videocrafter1 [5]; Zscope: Zeroscope; St. V. D.: Stable Video Diffusion [4]; St.Diff.: Stable Diffusion [42]; St. V.:Stable Video]

**Integration of Spatial and Temporal Information** Same as the previous work [3, 35] that utilizes ViT to learn video-level information, we transform each input video frame (*i.e.*, $\mathbf{x}$) into $L$ tokens, and propose IAFA for information aggregation, as depicted in Fig. 5a. Let us denote frame-level tokens as Patch-wise tokens (P-tokens), as they represent local information of one patch from $\mathbf{x}$. More formally, the $i$th frame, $\mathbf{x}_i$ is divided into $L$ patches, and all patches are projected into $\mathbf{T}_i = \{\mathbf{t}_i^j\}_{j=1}^L \in \mathbb{R}^{L \times D}$, where $D$ represents the dimension of each P-token. Also, to capture the global forgery information for each video frame, we introduce additional tokens called Frame-Centric tokens (FC-tokens). The FC-token is denoted as $\mathbf{p}_i \in \mathcal{R}^D$ for frame $\mathbf{x}_i$ and attends other tokens *within* the same frame $\mathbf{x}_i$.

During the forward propagation, we conduct IAFA based on a Transformer, with each block containing two self-attentions and consecutively modeling the local and global forgeries at each video frame. Specifically, the first self-attention captures dependencies among P-tokens that restore local forgery clues. This is formulated by Eq. 3 that $\mathbf{t}_i^j \in \mathbb{R}^D$ attends the token $\mathbf{t}_q^p \in \mathbb{R}^D$ that represents $p$th token from $q$th frame $\mathbf{x}_q$. Consequently, given the $i$th frame *i.e.*, $\mathbf{x}_i$, the second self-attention is conducted among the FC-token (*i.e.*, $\{\mathbf{p}_i\}$) and P-tokens (*i.e.*, $\{\mathbf{t}_i^0, \mathbf{t}_i^1, ... \mathbf{t}_i^{L-1}\}$) from the same frame, which encapsulates patch-wise forgery information into the global one for learning the more robust representation. We formulate this procedure in Eq. 4.

$$\mathbf{t}_i^j = \sum \mathrm{ATTN}(\mathbf{t}_i^j, \mathbf{t}_q^p) \quad i, j \in [1, N], j, p \in [1, L], \tag{3}$$

$$\mathbf{p}_i = \sum \mathrm{ATTN}(\mathbf{p}_i, \mathbf{t}_i^j) \quad j \in [1, L], \tag{4}$$

where $\mathrm{ATTN}$ refers to the self-attention operation.

### 3.3 Dynamic Fusion

We devise the dynamic fusion strategy (*i.e.*, $\mathcal{D}_f$) that combines spatiotemporal information from ST branch and MMFR (*i.e.*, $\mathbf{F}_f$ and $\mathbf{F}_m$) for the final prediction, by adjusting their contributions based on forgeries from the input. More formally, $\mathcal{D}_f$ (Fig. 5b) learns channel-wise dependencies among forgery representations (*e.g.*, $\mathbf{F}_{ST}$ and $\mathbf{F}_M$) via the attention mechanism, generating $\mathbf{w} \in \mathbb{R}^{N+M}$ as the output. This procedure can be expressed as $\mathbf{w} = \mathcal{D}_f(\mathrm{CONCAT}\{\mathbf{F}_{ST}, \mathbf{F}_M\})$. Also, $\mathbf{w}$ contains $\mathbf{w}_{ST} \in \mathbb{R}^N$ and $\mathbf{w}_M \in \mathbb{R}^M$, representing learned channel-wise weights for $\mathbf{F}_{ST}$ and $\mathbf{F}_M$, respectively. Such channel-wise weights are important in the fusion purpose, as they help emphasize useful information — we use $\mathbf{w}_{ST}$ and $\mathbf{w}_M$ to refine forgery representations as $\mathbf{F}'_{ST} = \mathbf{F}_{ST}\mathbf{w}_{ST}$ and $\mathbf{F}'_M = \mathbf{F}_M\mathbf{w}_M$. Lastly, $\mathbf{F}'_{ST}$ and $\mathbf{F}'_M$ are concatenated into the fused representation $\mathbf{F}_0 \in \mathbb{R}^{(M+N) \times D}$, which is used for the final scalar prediction $s$ via the average pooling (*i.e.*, $\mathrm{AVG}$) and linear layers(*i.e.*, $\mathrm{PROJ}$):

$$s = \mathrm{PROJ}(\mathrm{AVG}(\mathbf{F}_0)) = \mathrm{PROJ}(\mathrm{AVG}(\mathrm{CONCAT}\{\mathbf{F}'_{ST}, \mathbf{F}'_M\}). \tag{5}$$

## 4 Diffusion Video Forensics (DVF) Dataset

We construct a large-scale dataset for the video forensic task named Diffusion Video Forensics (DVF), as shown in Fig. 6. DVF contains 8 diffusion generative methods, including Stable Diffusion [42], VideoCrafter1 [5], Zeroscope, Sora, Pika, OpenSora, Stable Video, and Stable Video Diffusion[4].

To efficiently streamline the collection, we construct an effective automated pipeline that generates forgery videos based on real videos and prompts. Specifically, we start from two real video datasets,

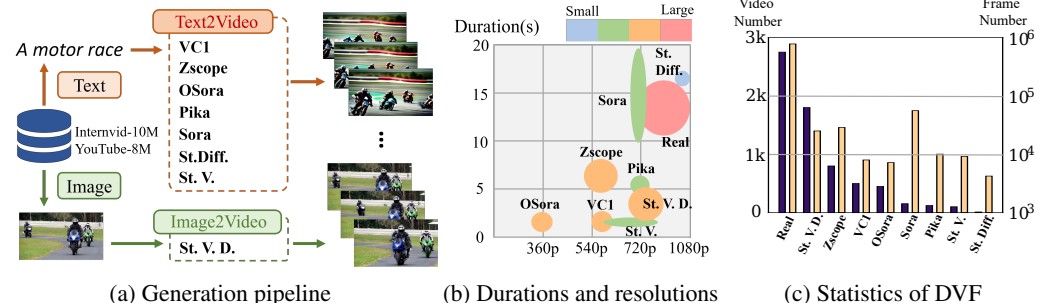

|  (a) Generation pipeline | (b) Durations and resolutions | (c) Statistics of DVF |

Figure 7: The overview of DVF dataset: (a) The procedure of forged video generation and collection. Real frames and captions sampled from Internvid-10M [54] and Youtube-8M [1] for text-to-video and image-to-video generation. (b) DVF contains videos at various resolutions and durations. (c) The statistics of DVF, measured by the numbers of frames and videos.[Key: VC1: Videocrafter1 [5]; Zscope: Zeroscope; OSora: OpenSora; St.Diff.: Stable Diffusion [42]; St. V.:Stable Video; St. V. D.: Stable Video Diffusion [4]]

Internvid-10M [54] and Youtube-8M [1]. Real videos are sampled for rich semantic content, with their frames and captions used for generation. Fig. 7 a introduces the generation process of DVF. For open-sourced generation methods, a prompt is fed to a text-to-video method(*i.e.* VideoCrafter1, Zeroscope, OpenSora), or a frame is provided to an image-to-video method(Stable Video Diffusion) to generate the corresponding fake video. For commercial and close-sourced datasets(*i.e.* Stable Diffusion, Stable Video, Sora, Pika), forgery videos are collected from official websites and social media. In total, we collect $3,938$ fake videos and $2,750$ real videos in DVF. As shown in Fig. 7 b and Fig. 7 c, our dataset contains multiple resolutions and durations. The video number of each dataset varies from $0.1k$ to $2.8k$, with the corresponding frame numbers from 4.2k to 784k. More details about DVF are provided in Appendix 8.3.1.

## 5 Training Strategy

This section details our two-stage training strategy, in which we first finetune the LMM branch via instruction tuning [28] and then optimize the entire framework in an end-to-end manner.

**LMM Branch Instruction Tuning** We first adapt LLaVA [28] to the forgery detection downstream task based on instruction tuning, an empirically effective way for various downstream tasks, which leverages LoRA [21] to improve the reasoning ability of Large Language Models(LLMs). For that, construct a large image-text paired dataset, named Rich Forgery Reasoning Dataset. Please refer to Appendix 8.3.2 for more details. We use multi-turn conversations to fine-tune the LMM, enhancing its ability to identify and judge the authenticity of input images. Following the instruction tuning strategy of LLaVA [28], we only fine-tune the projection layers and LLM in LLaVA. More formally, we formulate the objective function as the loss for an auto-regressive model, which is based on answer tokens from the LLM, as:

$$\mathcal{L}(\theta_1) = -\sum_{t=1}^{T} log(p_{\theta_1}(s^t|s^{i<t})), \tag{6}$$

where $s^i$ refers to the $i^{th}$ prediction token, $T$ refers to the length of total prediction tokens, and $\theta_1$ refers to the trainable parameters in the LMM.

**End-to-End Training** After fine-tuning LLaVA, we use this model to form LMM branch of MM-Det, and then the entire model is trained in an end-to-end manner. Please note that all parameters in LMM branch are frozen to ensure the optimal multi-modal representation can be obtained. More formally, we denote MM-Det's final prediction scalar and the ground truth as $s$ and $y$, respectively, and the model is optimized by the cross-entropy loss $\mathcal{L}$ as follows:

$$\mathcal{L}(\theta_2) = -(y \log \mathcal{D}(v) + (1-y) \log(1-\mathcal{D}(v)) \tag{7}$$

where $\theta_2$ refers to trainable parameters in both ST branch and dynamic fusion modules.

Table 1: Video forgery detection performance on the DVF dataset measured by AUC (%). [Key: **Best**; Second Best; Stable Diff.: Stable Diffusion; Avg.: Average]

| Method | Video-Crafter1 | Zero-scope | Open-Sora | Sora | Pika | Stable Diff. | Stable Video | Avg. |
|---|---|---|---|---|---|---|---|---|
| CNNDet [53] | $87.4_{\pm 1.5}$ | $88.2_{\pm 1.5}$ | $78.0_{\pm 1.6}$ | $63.8_{\pm 3.5}$ | $77.3_{\pm 2.1}$ | $73.5_{\pm 2.4}$ | $78.9_{\pm 4.1}$ | $78.2_{\pm 1.3}$ |
| DIRE [55] | $55.9_{\pm 2.2}$ | $61.8_{\pm 3.3}$ | $53.8_{\pm 1.8}$ | $60.5_{\pm 5.5}$ | $65.8_{\pm 1.7}$ | $62.7_{\pm 3.6}$ | $69.9_{\pm 2.5}$ | $62.1_{\pm 1.8}$ |
| Raising [8] | $63.8_{\pm 1.6}$ | $60.7_{\pm 1.9}$ | $64.1_{\pm 1.9}$ | $68.8_{\pm 3.6}$ | $70.7_{\pm 1.4}$ | $78.2_{\pm 2.3}$ | $62.8_{\pm 1.9}$ | $67.0_{\pm 0.9}$ |
| Uni-FD [36] | $75.0_{\pm 3.0}$ | $71.2_{\pm 3.4}$ | $76.6_{\pm 2.2}$ | $73.1_{\pm 1.5}$ | $76.2_{\pm 2.4}$ | $80.2_{\pm 1.9}$ | $66.7_{\pm 2.6}$ | $74.1_{\pm 1.2}$ |
| F3Net [38] | $89.7_{\pm 1.8}$ | $80.5_{\pm 2.2}$ | $69.3_{\pm 1.8}$ | $70.8_{\pm 6.9}$ | $88.9_{\pm 2.3}$ | $84.4_{\pm 2.1}$ | $85.1_{\pm 1.3}$ | $81.3_{\pm 1.9}$ |
| ViViT [3] | $79.1_{\pm 3.1}$ | $78.4_{\pm 2.0}$ | $77.7_{\pm 2.3}$ | $69.4_{\pm 3.5}$ | $83.1_{\pm 2.6}$ | $82.1_{\pm 2.0}$ | $83.6_{\pm 2.1}$ | $79.1_{\pm 1.8}$ |
| TALL [62] | $76.0_{\pm 1.4}$ | $65.9_{\pm 1.6}$ | $62.1_{\pm 1.3}$ | $64.3_{\pm 1.9}$ | $72.3_{\pm 2.9}$ | $65.8_{\pm 2.8}$ | $79.8_{\pm 2.2}$ | $69.5_{\pm 1.4}$ |
| TS2-Net [31] | $61.8_{\pm 3.9}$ | $70.6_{\pm 2.8}$ | $75.5_{\pm 3.4}$ | $78.0_{\pm 2.9}$ | $78.2_{\pm 2.8}$ | $62.1_{\pm 3.8}$ | $78.6_{\pm 3.0}$ | $72.1_{\pm 2.8}$ |
| DE-FAKE [43] | $74.7_{\pm 1.7}$ | $68.2_{\pm 2.9}$ | $55.8_{\pm 3.6}$ | $64.1_{\pm 3.1}$ | $85.6_{\pm 2.2}$ | $85.4_{\pm 2.6}$ | $70.6_{\pm 1.9}$ | $72.1_{\pm 2.2}$ |
| HiFi-Net [18] | $90.2_{\pm 3.0}$ | $89.7_{\pm 2.9}$ | $80.1_{\pm 2.6}$ | $70.1_{\pm 3.8}$ | $87.8_{\pm 2.9}$ | $89.2_{\pm 2.5}$ | $83.1_{\pm 2.2}$ | $84.3_{\pm 2.4}$ |
| MM-Det (Ours) | $\mathbf{93.5}_{\pm 3.6}$ | $\mathbf{94.0}_{\pm 2.8}$ | $\mathbf{88.8}_{\pm 2.8}$ | $\mathbf{86.2}_{\pm 1.8}$ | $\mathbf{95.9}_{\pm 2.8}$ | $\mathbf{95.7}_{\pm 2.5}$ | $\mathbf{89.9}_{\pm 2.0}$ | $\mathbf{92.0}_{\pm 2.6}$ |

# 6 Experiments

## 6.1 Setup

In the experiment, we use the proposed DVF for the evaluation. In training, $1,000$ videos from YouTube and $1,973$ fake videos generated by Stable Video Diffusion serve as the training set, in which $90\%$ are used for training and the remaining $10\%$ for validation. Real videos from Internvid-10M [54] and fake videos from 6 generative methods are used as testing samples. More details on training and testing are provided in Appendix 8.4.

For a fair comparison, we choose the following 10 recent detection methods as baselines. CNNDet [53] applies a ResNet [20] as the backbone for forgery detection. F3Net [38] utilizes frequency traces left in forgery content. HiFi-Net [18] devise a specific hierarchical fine-grained learning scheme to learn a wide range of forgery traces. Clip-Raising [9], Uni-FD [36] takes advantage of a pre-trained CLIP [39] as a training-free feature space. DIRE [55] detects diffusion images based on a reconstruction process of DDIM [44]. ViViT [3], TALL [62], and TS2-Net [31] take advantage of spatiotemporal information in various visual tasks. DE-FAKE [43] adopts visual and textual representations based on a CLIP encoder for image forgery detection. For the measurement, we choose AUC since it is a threshold-independent metric. We computer means and deviations of the performance across 5 runs on different random seeds.

## 6.2 Video Forgery Detection Performance

In Tab. 1, our proposed MM-Det achieves SoTA performance in detecting diffusion video, surpassing the second-best method, *i.e.*, HiFi-Net, by $7.7\%$ in the average of AUC scores. Specifically, for prior methods that are based on pre-trained CLIP features, such as Raising [8] and Universal FD [36], they remain effective on certain types of diffusion content (*i.e.* Stable Diffusion), but fail on most others. Simple structures like CNN [53] exceed these CLIP-based methods after being fine-tuned on our proposed DVF, reaching an average AUC score of $78.2\%$, which proves the necessity of such datasets. As for our method, MM-Det outperforms other methods in most datasets. Compared with frequency-based forgery methods, *e.g.*, HiFi-Net [38] and CLIP-based methods [8, 36], our method improves the performance from $+3.3\%$(VideoCrafter1) to $+15.5\%$(Stable Diffusion). It is worth mentioning that HiFi-Net makes the second-best performer in our DVF dataset, achieving $84.3\%$ AUC scores. We believe this indicates the multi-branch feature extractor used in HiFi-Net carries versatile forgery traces at multiple resolutions, enhancing the learning of the forgery invariant. The failure of frequency traces and CLIP features raises the need for more effective features. As for spatiotemporal baselines [3, 62, 31], we outperform them by $+12.9\%$(ViViT), $+22.5\%$(TALL) and $+19.9\%$(TS2-Net), demonstrating the effective features of MMFR and IAFA. At last, our detector improves by $19.9\%$ to another multi-modal detector [57], which utilizes visual information and corresponding captions for feature enhancement. It is shown that the introduction of MMFR is more generalizable than a simple combination of visual features and text descriptions in that the powerful perceptual and reasoning abilities of LMMs play a crucial role in discriminating between real and fake content.

Table 2: Ablation analysis measured by AUC (%). [Key: **Best**; Avg.: Average; Rec.: Diffusion Reconstruction Procedure; Fus. Dynamic Fusion Strategy].

| | | Modules | | | Video-Crafter1 | Zero scope | Open Sora | Sora | Pika | Stable Diff. | Stable Video | Avg. |
|---|---|---|---|---|---|---|---|---|---|---|---|---|
| ViT | Rec. | IAFA | MMFR | Fus. | | | | | | | | |
| ✔ | | | | | $68.2_{\pm1.3}$ | $80.1_{\pm2.8}$ | $64.8_{\pm2.1}$ | $59.6_{\pm1.9}$ | $79.2_{\pm2.9}$ | $89.2_{\pm2.6}$ | $87.4_{\pm2.1}$ | $75.5_{\pm1.8}$ |
| ✔ | ✔ | | | | $70.1_{\pm1.6}$ | $76.3_{\pm1.9}$ | $77.6_{\pm1.8}$ | $66.2_{\pm2.0}$ | $80.2_{\pm1.9}$ | $83.9_{\pm2.5}$ | $86.1_{\pm2.1}$ | $77.2_{\pm1.3}$ |
| ✔ | ✔ | ✔ | | | $90.2_{\pm2.8}$ | $90.1_{\pm2.5}$ | $85.8_{\pm1.8}$ | $82.0_{\pm2.8}$ | $93.9_{\pm1.8}$ | $91.9_{\pm2.6}$ | $85.7_{\pm2.7}$ | $88.5_{\pm1.7}$ |
| | | | ✔ | | $89.7_{\pm3.8}$ | $93.4_{\pm3.2}$ | $84.6_{\pm3.2}$ | $83.2_{\pm2.6}$ | $94.0_{\pm2.2}$ | $89.7_{\pm2.0}$ | $86.2_{\pm2.8}$ | $88.7_{\pm2.8}$ |
| ✔ | ✔ | ✔ | ✔ | | $92.1_{\pm3.1}$ | $93.8_{\pm2.9}$ | $86.5_{\pm2.6}$ | $83.2_{\pm1.9}$ | $90.1_{\pm2.5}$ | $90.2_{\pm2.1}$ | $87.6_{\pm2.9}$ | $89.1_{\pm2.3}$ |
| ✔ | ✔ | ✔ | ✔ | ✔ | $\mathbf{93.5}_{\pm3.6}$ | $\mathbf{94.0}_{\pm2.8}$ | $\mathbf{88.8}_{\pm2.8}$ | $\mathbf{86.2}_{\pm1.8}$ | $\mathbf{95.9}_{\pm2.8}$ | $\mathbf{95.7}_{\pm2.5}$ | $\mathbf{89.9}_{\pm2.0}$ | $\mathbf{92.0}_{\pm2.6}$ |

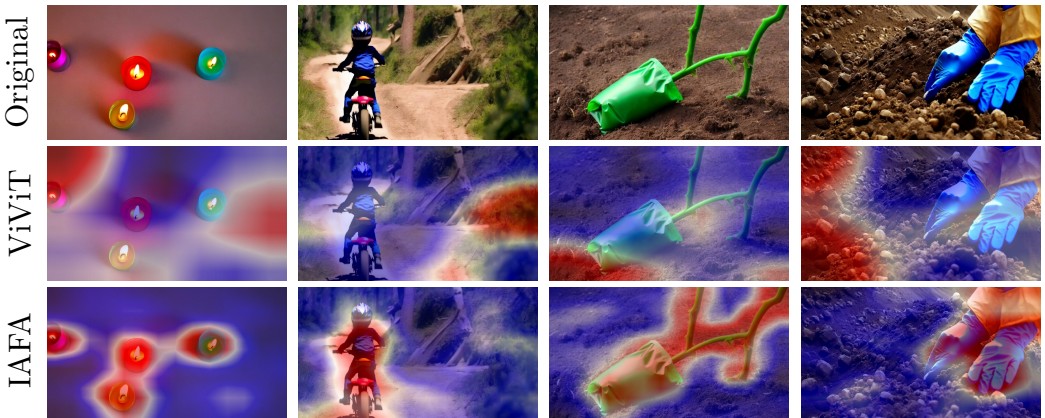

Figure 8: Visualization of artifacts captured from our IAFA and ViViT [3]. We use activation maps to highlight spatial weights within each frame. All content is generated by VideoCrafter1. Features from the last layer of transformers are extracted for visualization.

## 6.3 Ablation Study

Tab. 2 shows the impact of individual modules proposed in MM-Det. Specifically, we use the Hybrid ViT [11] as the base model and incorporate it with the reconstruction procedure for diffusion trace amplification, which enhances the detection performance by +1.7% AUC score. Such a module raises the performance in OpenSora and Sora, revealing that frequency-based information benefits forgery detection on these methods. Detection performance is further increased by using IAFA, which strengthens the learning between in-frame and cross-frame information, increasing a +13.0% AUC score to the base model. The rise in performance indicates such temporal information benefits most types of forgery video detection. After that, Tab. 2's line 4 indicates the effectiveness of MMFR: a detector purely based on such representation can receive 88.7% performance in AUC, +13.2% higher than the base model. In addition, by merging MMFR (*i.e.*, LMM branch) and ST branch, the performance rises by +0.4%. Finally, with the dynamic fusion strategy, our method receives an impressive 92.0% AUC score for all generative methods, higher than every single feature. These experiments highlight the necessities of each module in our framework. Moreover, an ablation study on LLMs is detailed in Appendix 8.5 to prove the effectiveness of various LLMs in MM-Det.

## 6.4 Spatio Temporal Information Anaylsis

In the analysis of IAFA, we visualize feature activation maps from the last layer of ViT in the ST branch based on the L2-norm. We compare our feature maps with another spatiotemporal baseline, ViViT [3], as depicted in Fig. 8. While attention maps of ViViT are sparse and irregular, the ones of our IAFA have a tendency to concentrate on the segmentation of diffusion-generated objects, indicating that IAFA captures typical spatial forgery regions in frames. The attention mainly focuses on common forgery traces, such as blurred generative patterns and defective parts of objects, signaling that diffusion models might find it difficult to generate delicate content. The concentration of activation on certain informative objects discloses both spatial artifacts of existing generative methods, demonstrating the effectiveness of our proposed ST branch.

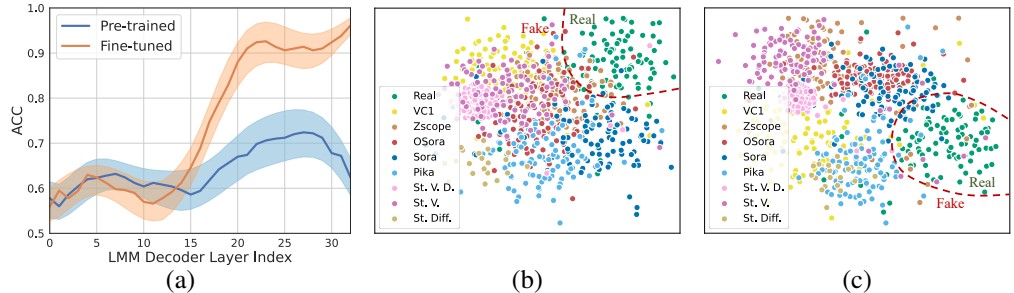

Figure 9: (a): Clustering accuracy using features from different layers in LMM branch showcases MMFR's ability to discern forgeries. (b)(c): t-SNE [52] visualization of features from ST and LMM branches. For each dataset, 100 videos are sampled and clustered for good visibility. Both features demonstrate boundaries between real and forgery videos. (b) Features from the ST branch. (c): Features from the LMM branch.

## 6.5 Multi-modal Forgery Representation Analysis

Fig. 9 details the effectiveness of MMFR in LMM branch. First, as depicted in Fig. 9 a, we quantify the detection ability of features from each Transformer-based decoder layer in the LLM. Specifically, we evaluate both pre-trained LLaVA [28] and its fine-tuned version on the task of distinguishing real and fake frames. These frames are randomly sampled from $1,000$ real videos and equivalent fake ones from Stable Video Diffusion [42]. Layer-wise outputs from the large language model in LLaVA (*i.e.*, Vicuna [50]) are obtained — for $i$ th layer in the LLM, we denote its output features as $\mathbf{f}_i^o$, $i \in [1, 33]$. The K-Means clustering algorithm is adopted to evaluate the classification accuracy based on $\mathbf{f}_i^o$. Empirically, we observe that features extracted from a fine-tuned LLaVA show promising classification accuracy in the last few layers, *e.g.*, 22-nd or later layers. This phenomenon indicates that specific layers in LLMs indeed generate features that can be used for image forensic tasks. Such features are utilized in MMFR to exhibit high generalization ability towards diverse and unseen forgeries. Secondly, the comparison between pre-trained and fine-tuned LLaVA highlights the importance of downstream task-oriented instruction tuning for LMMs. This conclusion is consistent with the findings of prior works [27, 28, 69, 68].

In addition, shown in Fig. 9 b and 9 c, we analyze features from ST branch and LMM branch through t-SNE [52]. Both features achieve superior performance in separating real and forgery videos. In Fig. 9 b, spatiotemporal information forms a rough boundary between real and fake videos. This feature is effective for VideoCrafter1 [5], Zeroscope, Stable Video, and Pika, whose durations and resolutions are similar to the training set. However, the detection performance might decrease on Sora and OpenSora with overlay in the clusters. We suppose that various resolutions and durations may compromise the generalization ability, magnifying the importance of a comprehensive dataset for these videos. Fig. 9 c demonstrates the more powerful feature from LMM branch. Samples from Zeroscope, Sora, and Pika are compacted into a denser area, indicating the ability of LLMs to conduct generalizable reasoning. Such features provide new insights for detection when spatial and temporal artifacts are not obvious among the latest forgery videos.

## 7 Conclusion

In this work, we develop an effective video-level algorithm termed Multi-Modal Detection (MM-Det) for diffusion-generated video detection. MM-Det leverages a novel generalizable Multi-Modal Forgery Representation (MMFR) that is obtained from multi-modal spaces in LMMs. Specifically, the proposed MM-Det has two major branches: the LMM branch, which incorporates vision and text features from the fine-tuned foundation model, and the ST branch, which concentrates on modeling spatial-temporal information aggregated through In-and-Across Frame Attention. Extensive experiments demonstrate the effectiveness of our proposed detector. In addition, we establish a comprehensive dataset for various diffusion generative videos, which we hope will serve as a general benchmark for real-world video forensic tasks.

**Acknowledgement** This work was supported in part by the National Natural Science Foundation of China under Grant 62301310 and 62225112, and in part by Sichuan Science and Technology Program under Grant 2024NSFSC1426.

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

# 8 Appendix / Supplemental Material

## 8.1 Limitations

Although our proposed MM-Det advances in detecting fake videos, further issues are left to handle. First, as the landscape of video manipulation technology evolves, new techniques and tools will outpace existing detection methods. The gap between training data and real-world applications can lead to misleading results. Currently, our method is limited to fully synthesized diffusion videos, lacking the generalization to more delicate forgeries like partial manipulation. A possible reason is that small forgery traces disappear after multiple downsampling operations in the deep network of LMMs. Besides, the integration of a large language model into detection costs huge computational complexity, which is not an optimal choice for an environment with limited resources.

In conclusion, while our algorithm makes a critical step forward in detecting fake videos, it faces significant challenges due to the rapid advancement of video manipulation technologies. Addressing these limitations requires further research to keep pace with the evolving techniques in digital content manipulation.

## 8.2 Broader Impacts

In this work, our team has developed an effective algorithm to detect fake videos, a breakthrough that promises to fortify the authenticity of online media. In real-world social media where misinformation can spread rapidly, our method acts as a crucial safeguard by empowering platforms to flag and remove deceptive videos before they can mislead users. However, our methods may fail in extreme situations, such as blurred images or noisy images. The algorithm should be carefully treated to avoid misleading results. The long-term influence of our work protects public trust by ensuring the authenticity of digital content. Meanwhile, precaution is needed for a fair application.

## 8.3 Datasets

### 8.3.1 Diffusion Video Forensics

We propose a comprehensive dataset, named Diffusion Video Forensics (DVF), for diffusion video forensics, as shown in Tab. S1. DVF consists of fake videos generated from 8 different generation methods, covering text-to-video and image-to-video generative methods. In total, We make a collection of $2,788$ real videos and $4,111$ fake videos. Real videos are from YouTube and Internvid-10M [54].

We formally introduce the generation pipeline for video collection. Generation methods in DVF are divided into closed-sourced methods and open-sourced methods. For closed-sourced methods(Sora, Pika, Stable Diffusion [42] and Stable Video), we collect video samples from official websites and social media like TikTok to form the forgery video datasets. For open-sourced methods, the generation pipelines are divided into text-to-video (OpenSora, VideoCrafter1 [5] and Zeroscope) and image-to-video (Stable Video Diffusion [4]). For a text-to-video generative method, real data derives from a text-image paired video dataset, Internvid-10M. Specifically, we fetch paired real videos $R = \{\mathbf{r}_1, \mathbf{r}_2, ..., \mathbf{r}_N\}$ and corresponding captions $C = \{c_1, c_2, ..., c_N\}$. We directly apply the

Table S1: Diffusion Video Forensics Composition [Key: T2V: Text-to-Video; I2V: Image-to-Video]

| Dataset | Source | Video Number | Resolution |
|---------|--------|--------------|------------|
| Real | Youtube & Internvid-10M | $2,750$ | $1,280 \times 720$ |
| Stable Video Diffusion | I2V | $1,800$ | $1,024 \times 576$ |
| VideoCrafter1 | T2V | $450$ | $1,024 \times 576$ |
| Zeroscope | T2V | $800$ | $1,024 \times 576$ |
| Sora | Social Media | $153$ | $1,280 \times 720$ |
| Pika | T2V, I2V | $122$ | $1,280 \times 720$ |
| OpenSora | T2V | $500$ | $512 \times 512$ |
| Stable Diffusion | I2V | $12$ | $1,080 \times 1,920$ |
| Stable Video | T2V | $101$ | $1,024 \times 576, 1,920 \times 1,080$ |

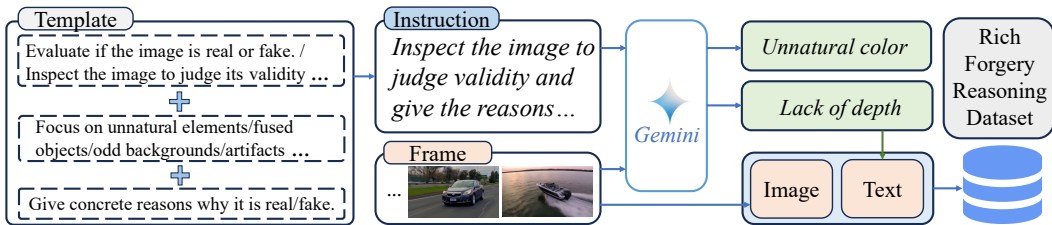

Figure S1: The overview of Rich Forgery Reasoning Dataset. To obtain text-image paired data on forgery reasoning, we take advantage of the powerful reasoning ability of Gemini [48] to generate ground truth for each image. First, an instruction $T$ is sampled from a template, along with a frame $\mathbf{f} \in \mathbb{R}^{H \times W \times C}$ fed into Gemini for forgery analysis and detection. The response $r$ contains detailed judgment and reasoning on the content and authenticity of $\mathbf{f}$ (*e.g.*, analyses on color and depth). Finally, $\mathbf{f}$ and $r$ are collected as the text-image paired data to form Rich Forgery Reasoning Dataset.

captions as prompts to generate fake video datasets $F$, such that $F = \{\mathbf{f}_1, \mathbf{f}_2, ...\mathbf{f}_N\}, \mathbf{f}_i = g_t(c_i), i \in [1, N]$, where $g_t$ denotes a text-to-video method. Both $R$ and $F$ are included in DVF as real and fake datasets. For image-to-video methods, real videos come from Youtube-8M [1], which are denoted as $R = \{\mathbf{r}_1, \mathbf{r}_2, ..., \mathbf{r}_N\}, \mathbf{r}_i \in \mathbb{R}^{L \times H \times W \times C}$. For each video $r_i$, a real frame $x_i \in \mathbb{R}^{H \times W \times C}$ is randomly sampled from $r_i$ and serves as the conditional input for generation. The fake datasets are obtained as $F = \{\mathbf{f}_1, \mathbf{f}_2, ...\mathbf{f}_N\}, \mathbf{f}_i = g_{im}(r_i), i \in [1, N]$, where $g_{im}$ denotes an image-to-video method.

### 8.3.2 Rich Forgery Reasoning Dataset

We construct a text-image paired dataset, called Rich Forgery Reasoning Dataset (RFRD), to support instruction-tuning LMMs on the forgery detection task, as shown in Fig. S1. We start from the YouTube and Stable Video Diffusion dataset in DVF, where we select $1,000$ real videos and $1,800$ fake videos. Depending on the powerful reasoning ability of Gemini [48] v1.5 Pro, we follow the scheme in Fig. S1 to generate ground truth for frames. In total, $1,921$ real frames and $3,579$ fake frames are sampled to generate $5,500$ image-text paired textual descriptions. These descriptions are then cleaned and converted into 38k multi-turn conversations for fine-tuning LLaVA [28] in LMM branch of MM-Det.

## 8.4 Implementation Details

**Hyperparameters of MM-Det**   We introduce the implementation of our MM-Det. In ST branch, we employ a Hybrid-ViT [11] to build the video-level feature encoder. We choose Hybrid-ViT-B, with a ViT-B/16 on top of a ResNet-50 backbone, the patch size $14 \times 14$, and the hidden size $768$. We employ IAFA in each attention block of the ViT, with FC-tokens initialized as the class token of ViT. In the embedding stage, learnable spatial and temporal embeddings are introduced in the form of addition. All patches at the same position within frames share the same spatial embedding and patches with the same timestep share the same time embedding. A pre-trained VQ-VAE is applied to reconstruct videos, with hidden size $d = 256$ and the codebook size $K = 512$. We train the VQ-VAE on $50,000$ images from ImageNet. In LMM branch, we utilize LLaVA [28] v1.5, with a CLIP [39] encoder $\mathcal{E}$ of CLIP-ViT-L-patch14-336 and a large language model $\mathcal{D}$ of Vicuna-7b for reasoning. Our proposed MMFR is composed of the pooler output $\mathbf{F}_V \in \mathbb{R}^{1024}$ from the last layer of $\mathcal{E}$ and the embedding of output $\mathbf{F}_L$ from the last layer of $\mathcal{D}$. To balance effectiveness and efficiency, we fix the length of output tokens into $64$ and obtain $\mathbf{F}_L \in \mathbb{R}^{64 \times 4096}$.

**Training and Inference**   As for the experimental resources in training and inference, we conduct all experiments using a single NVIDIA RTX 3090 GPU and a maximum of 200G memory.

The training strategy of MM-Det is in two-stage. We first conduct instruction tuning for LLaVA in the LMM branch based on LoRA [21]. We start from a pre-trained LLaVA v1.5 and train it on our collected Rich Forgery Reasoning Dataset detailed in Sec. 8.3.2. We use an Adam optimizer with the learning rate set as $2e^{-5}$ for 10 epochs. After that, we integrate LLaVA into MM-Det and conduct the overall training. The training set is split into $9 : 1$ for training and validation data. For each video,

Table S2: Evaluation on different Large Language Models measured by AUC(%). [Key: **Best**; Avg.: Average].

| LLM | Video-Crafter1 | Zeroscope | OpenSora | Sora | Pika | Stable Diffusion | Stable Video | Avg. |
|---|---|---|---|---|---|---|---|---|
| N/A | $90.2_{\pm2.8}$ | $90.1_{\pm2.5}$ | $85.8_{\pm1.8}$ | $82.0_{\pm2.8}$ | $93.9_{\pm1.8}$ | $91.9_{\pm2.6}$ | $85.7_{\pm2.7}$ | $88.5_{\pm1.7}$ |
| Vicuna-7b | $93.5_{\pm3.6}$ | $\mathbf{94.0}_{\pm2.8}$ | $88.8_{\pm2.8}$ | $\mathbf{86.2}_{\pm1.8}$ | $95.9_{\pm2.8}$ | $95.7_{\pm2.5}$ | $89.9_{\pm2.0}$ | $92.0_{\pm2.6}$ |
| Vicuna-13b | $\mathbf{95.5}_{\pm2.5}$ | $93.2_{\pm2.2}$ | $\mathbf{89.2}_{\pm3.1}$ | $83.9_{\pm2.5}$ | $\mathbf{96.6}_{\pm1.9}$ | $\mathbf{95.9}_{\pm2.3}$ | $\mathbf{90.8}_{\pm2.7}$ | $\mathbf{92.2}_{\pm2.4}$ |
| Mistral-7b | $92.6_{\pm2.9}$ | $92.1_{\pm2.5}$ | $86.3_{\pm3.6}$ | $83.2_{\pm2.2}$ | $94.6_{\pm2.6}$ | $93.6_{\pm2.0}$ | $86.2_{\pm2.8}$ | $89.8_{\pm2.6}$ |

Table S3: Perfomance of MM-Det on common post-processing operations measured by AUC(%).

| N/A | Blur $\sigma = 3$ | JPEG $Q = 50$ | Resize 0.7 | Rotate 90 | Mixed |
|---|---|---|---|---|---|
| 92.0 | 86.2 | 91.1 | 89.9 | 90.1 | 88.6 |

successive 10 frames are randomly sampled and cropped into $224 \times 224$ as the input. We use an Adam optimizer with the learning rate set as $1e^{-4}$ for training until the model converges.

For inference, we evaluate all models at the video level. For frame-level baselines, the score of an entire video is obtained as the average score of all frames. For video-level methods, successive clips are fed according to the corresponding window size, and the entire score is obtained as the average score of all clips. In addition, to leverage the efficiency for inference, MM-Det first caches MMFR for each video by conducting reasoning at the interval of 1 frame every 6 seconds. During inference, each video clip directly applies MMFR from the nearest cached frame as an approximation to reduce the huge computational cost. For each baseline, we conduct evaluations on 5 different seeds $(1, 100, 999, 1234, 9999)$ and choose the average score.

## 8.5 Ablation Study on LLMs

We adopt alternative LLMs in our MM-Det to evaluate different choices of language backbones. Performance is reported in Tab. S2. More formally, for a fair comparison, when using different LLMs, we maintain other components *e.g.*, CLIP, the reconstruction procedure, IAFA, and the dynamic fusion of the original MM-Det remained. Specifically, the introduction of a combined vision and text space from Vicuna-7b in LLaVA improves the performance by $+3.5\%$. As for the choice of LMMs, Vicuna-7b achieves an average AUC score of $92.0\%$, $+2.2\%$ higher than Mistral-7b. We suppose this result may be attributed to different attention mechanisms in Vicuna and Mistral. Vicuna-13b gains a further improvement by $+0.2\%$ due to incremental parameters in capturing more effective multi-modal feature spaces. These results prove that our MMFR is effective and extensible to other language models.

## 8.6 Robustness Analysis

To analyze the robustness of our method, we conduct an additional evaluation of MM-Det based on common post-processing operations. We choose Gaussian blur with $\sigma = 3$, JPEG compression with quality $Q = 90$, resize with a ratio of $0.7$, rotation with an angle of $90$, and a mixture of all operations as unseen perturbations in real-world scenarios. Testing samples are selected from DVF to form a total of $500$ real videos and $500$ fake videos. As reported in Tab. S3, MM-Det meets a degradation of $0.9\%$(JPEG Compression) to $5.8\%$(Gaussian blur), with all performance above $86\%$. The results indicate the effectiveness of our method under these operations.

