# OpenReview forum: "On Learning Multi-Modal Forgery Representation for Diffusion Generated Video Detection"
_NeurIPS.cc/2024/Conference — NeurIPS 2024 poster_

### Official Review · Reviewer_Niir · 2024-06-17

**Soundness:** 2
**Presentation:** 3
**Contribution:** 3
**Rating:** 6
**Confidence:** 4

**Summary:**

To detect video forensics, the authors propose an innovative Multi-Modal Forgery Representation (MMFR) to discriminate fake videos from real ones. Besides, the authors establish a high-quality dataset including videos generated from various diffusion-based algorithems to evaluate the effectiness of the proposed detector.

**Strengths:**

（1）The authors propose a video-level detection algorithm, named MM-Det, to capture forgery traces based on an LMM-based representation for the generalizable fake video detection. MM-Det utilizes perception and reasoning capabilities from LMMs through multi-turn conversations to learn a generalizable forgery feature.
（2）The authors extent reconstruction representation for detecting diffusion images into the video domain to amplify diffusion artifacts both in spatial and temporal information.
（3）The authors are the first to establish a comprehensive dataset for diffusion-generated videos, named DVF. The proposed detector achieves promising detection performance on a wide spectrum of forgery videos.

**Weaknesses:**

(1) I didn't understand how the authors use VQVAE to amplify diffusion features, and it seems that this reconstruction branch is not shown in Figure 3.
(2) There have been some works [1,2] that incorporate temporal token in self-attention, is this different from IAFA? What are the advantages of the proposed method?
[1] Wang J, Yang X, Li H, et al. Efficient video transformers with spatial-temporal token selection[C]//European Conference on Computer Vision. Cham. Springer Nature Switzerland, 2022: 69-86.
[2] Liu Y, Xiong P, Xu L, et al. Ts2-net: Token shift and selection transformer for text-video retrieval[C]//European conference on computer vision. Cham: Springer Nature Switzerland, 2022: 319-335.
(3) As shown in Section 4.2, the authors collected a training set and a test set to evaluate the performance of different methods. What is the relationship between these data and DVF? The article does not seem to describe the benefits of the DVF, e.g., how much performance improvement does the DVF bring to the authors' method, and how much performance improvement does the DVF bring to other methods?
(4) Forgery video detection is challenging task, however, the authors' experimental content is too small and simple, only 2 tables of experimental content is difficult to fully verify the effectiveness of the proposed method. The authors need to do some other types of experiments to further validate the performance of the method.

**Questions:**

From the visualization results, the colors of the fake image are more colorful, is the author's method still valid if the brightness is adjusted to make them similar to the real image?

**Limitations:**

Yes.

---

> ### Author Rebuttal · Authors · 2024-08-05
>
> We thank the reviewer for the careful and helpful feedback. We appreciate all constructive comments on the novelty, clarity, and valuable contribution of the paper. We demonstrate additional experiments and answer all weaknesses and questions mentioned in the review.
> > **Q1** The usage of VQVAE
>
> As for diffusion detection, a series of previous works[1,2] found that the reconstruction process of autoencoders extracts discriminative features for diffusion. Following [2], our work introduces a pretrained vector quantized variational autoencoder (VQVAE) for an augmentation to diffusion forgery features. We display the contribution of VQVAE reconstruction in Figure B of our PDF material.
>
> For brevity and legibility, we do not visualize an explicit reconstruction process in Figure 3 of our paper, as it is not the main contribution of our method. We have provided a detailed explanation of this process in Section 3.2, Lines 147-158 of our paper.
>
> > **Q2** Analysis of IAFA
>
> We provide analysis and experiments in Q1 of the overall response.
>
> > **Q3**: The relationship between experiment datasets and our proposed DVF
>
> Both training and testing datasets are subsets of DVF. They do not overlap with each other and form the entire DVF together.
>
> Specifically, the dataset DVF includes real videos from Internvid-10M and fake videos from 8 diffusion methods. In experiments, we choose one diffusion method, Stable Video Diffusion, as the training sets, and evaluate all baselines on the other 7 diffusion methods.
>
> > **Q4**: Contribution of DVF
>
> We provide additional experiments to prove the contribution of DVF on performance in Table I. For our method, we emphasize the contribution of DVF in finetuning LMMs. For other methods, DVF provides diffusion traces for forgery detection. We prove the effectiveness of DVF by comparing pretrained baselines with ones fine-tuned on DVF, for which we choose UnviersalFD[5] and F3Net[6], and comparing our method between pretrained and fine-tuned LMMs.
>
> **Table I**: Performance of pretrained and fine-tuned detectors on DVF in terms of AUC
> |Model|Videocrafter|Zeroscope|Sora|Pika|Opensora|Stable Diffusion|Stable Video|Average|
> |:--|:--:|:--:|:--:|:--:|:--:|:--:|:--:|:--:|
> |UniversalFD(Pretrained)|$59.2$|$69.4$|$65.6$|$65.2$|$61.7|$81.2$|$59.2$|$65.9$|
> |UniversalFD(Finetuned)|$93.6$|$90.1$|$85.4$|$93.0$|$83.9$|$81.5$|$87.9$|$87.9$|
> |F3Net(Pretrained)|$60.1$|$65.2$|$60.7$|$63.6$|$59.2$|$65.8$|$61.3$|$62.3$|
> |F3Net(Finetuned)|$96.1$|$91.8$|$66.0$|$95.6$|$85.9$|$86.3$|$96.0$|$88.2$|
> |Ours(w/Pretrained LMM)|$95.3$|$95.2$|$89.6$|$94.3$|$90.2$|$86.6$|$96.7$|$92.6$|
> |Ours(w/Finetuned LMM)|$99.2$|$98.4$|$97.4$|$95.5$|$99.4$|$98.0$|$98.4$|$98.0$|
>
> As is shown, DVF benefits other baselines in $+22.0\\%$ and $+25.9\\%$ in AUC. For our method, the LMM finetuned on DVF also outperforms pretrained one by $+5.4\\%$ in AUC.
>
> > **Q5**: More experiments on validation of our method
>
> We provide further analysis and experiments to validate our method. In the overall response, we provide
>
> 1) the functionality of IAFA compared with other baselines in Q1,
>
> 2) the generalization of our method on GAN and latest diffusion methods in Q2,
>
> 3) the performance of our method on different video durations and resolutions in Q2,
>
> 4) the ablation study on different LLMs in Q3.
>
> In our PDF material, we provide
>
> 1) Figure A: attention heatmaps of our proposed IAFA compared with other spatiotemporal networks.
>
> > **Q6**: From the visualization results, the colors of the fake image are more colorful, is the author's method still valid if the brightness is adjusted to make them similar to the real image?
>
> Our method is effective after adjusting the brightness. We provide more visualization results of our LMMs on both real and fake content with similar brightness in Figure C of our PDF material.
>
> **References**
>
> [1] Wang Z, Bao J, Zhou W, et al. Dire for diffusion-generated image detection
>
> [2] Ricker J, Lukovnikov D, Fischer A. AEROBLADE: Training-Free Detection of Latent Diffusion Images Using Autoencoder Reconstruction Error
>
> [3] Wang J, Yang X, Li H, et al. Efficient video transformers with spatial-temporal token selection
>
> [4] Liu Y, Xiong P, Xu L, et al. Ts2-net: Token shift and selection transformer for text-video retrieval
>
> [5] Ojha U, Li Y, Lee Y J. Towards universal fake image detectors that generalize across generative models
>
> [6] Qian Y, Yin G, Sheng L, et al. Thinking in frequency: Face forgery detection by mining frequency-aware clues

---

> > ### Comment · Reviewer_Niir · 2024-08-13
> >
> > The author did additional experiments to solve my concerns, so I have increased my score to 6 (weak accept).

---

> ### Author Response · Authors · 2024-08-13
> **Thank you, Reviewer Niir**
>
> Dear Reviewer Niir,
>
> We are glad that our additional experiments solved your concerns. Thanks again for your valuable review.
>
> Best regards,
>
> Authors of Submission 2768

---

### Official Review · Reviewer_LBha · 2024-07-12

**Soundness:** 4
**Presentation:** 3
**Contribution:** 4
**Rating:** 7
**Confidence:** 5

**Summary:**

This work focuses on diffusion model detection, a core and popular research topic recently. It identifies limitations in previous studies that concentrate on fake face and image-level detection and explores the idea of using recent LMM to detect forgery. The proposed Multi-modal Forgery Representation (MMFR) leverages existing LMM and introduces a new in-and-across frame attention mechanism. Additionally, a new dataset is proposed, and empirical results demonstrate the effectiveness of the proposed methods.

**Strengths:**

1. The motivation behind this work is clear, and the idea of using LMMs for image forensics is novel. This idea potential pioneers a new research direction.

2. Constructing a large-scale diffusion video dataset for the community is a reasonable endeavor and can benefit the entire research community.

3. I personally appreciate the analysis in section 4.4, which explains how the authors designed the MMFR and utilized outputs from specific layers of LLama.

4. The main result suggests the proposed method achieves SoTA video-level performance and can be used as the baseline method for the future works.

**Weaknesses:**

1. To me, the contribution of in-and-across frame attention seems limited. Providing a comparison to other video-level ViTs and more details about its differences from other works regarding ViT would make this contribution more convincing.

2. The DE-FAKE [R1] method also adopts multi-modal representations, but this work is neither discussed nor compared in the paper.

 [R1: DE-FAKE: Detection and Attribution of Fake Images Generated by Text-to-Image Generation Models]

3. The proposed dataset is a significant contribution of the paper. Therefore, instead of individual frames, it would be more convincing to include sample video frames either in the supplementary material or on the project page.

4. The main performance table needs more details and a clarification on why it contains both frame and video comparisons.

**Questions:**

Please refer to the weakness section.

1. The overall work is novel and demonstrates strong empirical performance, but I have minor concerns regarding its contributions, specifically the in-and-across attention mechanism and the dataset.

2. Additionally, more related work should be discussed and compared to provide a comprehensive context.

**Limitations:**

Yes, the authors adequately discussed the limitations and societal impact of their work in Appendix.

---

> ### Author Rebuttal · Authors · 2024-08-03
>
> We thank the reviewer for all insightful comments and suggestions. We appreciate the appraisal of the contribution, soundness, and clarity of the paper. We provide additional experiments and answer the weaknesses and questions in the review.
>
> > **Q1**: Comparison with other spatiotemporal networks
>
> We provide a comparison of IAFA with other ViT baselines and analyze our advantages in Q1 of the overall response.
>
> > **Q2**: Discussion and comparison with other multimodal baselines
>
> In previous studies on forgery detection, DEFAKE[1] adopts multimodal representation for forgery detection by using semantic descriptive prompts as an augmentation for visual features. However, semantic descriptions do not have strong correlations with the authenticity of images, which is ineffective in forgery tasks. As a comparison, our method utilizes the powerful reasoning ability of finetuned Large Multimodal Models (LMMs) to provide more accurate features in the text space. Therefore, our method is more effective in forgery detection.
>
> We provide a comparison of DEFAKE[1] with our method in Table H. Each model is trained on the same training dataset SVD, and evaluated on other diffusion datasets in our proposed dataset DVF.
>
> **Table H**: Comparison between DEFAKE and our method in AUC
> |Model|Videocrafter|Zeroscope|Sora|Pika|Opensora|Stable Diffusion|Stable Video|Average|
> |:--|:--:|:--:|:--:|:--:|:--:|:--:|:--:|:--:|
> |DEFAKE(SIGSAC2023)|$72.3$|$70.3$|$67.3$|$88.4$|$53.6$|$86.0$|$74.1$|$73.1$|
> |Ours|$99.2$|$98.4$|$97.4$|$95.5$|$99.4$|$98.0$|$98.4$|$98.0$|
>
> As is demonstrated, our proposed multimodal representation is effective in most diffusion forgery content.
>
> > **Q3**: Visualization of DVF
>
> We will release sample video frames from DVF instead of individual frames for better visualization.
>
> > **Q4**: A clarification of the main table
>
> To justify the effectiveness of our method both on the image-level and video-level detection, the main performance table contains both frame and video comparisons. For frame-level detection, frames from each video are treated individually for training and testing to demonstrate the generalization ability of our proposed Multi-Modal Forgery Representation(MMFR) without spatio-temporal information. For video-level detection, continuous video clips are forwarded during training and evaluation. In this case, both multimodal representation and spatiotemporal information take effect. We use these experiment settings to validate the effectiveness of MMFR and IAFA.
>
> > **Q5**: Discussion and comparison with more works
>
> We provide additional analysis and experiments regarding more related works.
>
> In the overall response, we provide
>
> 1) the functionality of IAFA compared with other baselines in Q1,
>
> 2) the generalization of our method on GAN and latest diffusion methods in Q2,
>
> 3) the performance of our method on different video durations and resolutions in Q2,
>
> 4) the ablation study on different LLMs for our method in Q3.
>
> In the PDF material, we provide
>
> 1) Figure A:  Visualization on attention heatmaps from IAFA.
>
> **References**
>
> [1] Sha Z, Li Z, Yu N, et al. De-fake: Detection and attribution of fake images generated by text-to-image generation models
>
> [2] Arnab A, Dehghani M, Heigold G, et al. Vivit: A video vision transformer

---

> > ### Comment · Reviewer_LBha · 2024-08-12
> >
> > Thank you for the clear rebuttal. My concerns have been addressed. I have increased my score to 7 (accept). It would be good to include some of clarifications provided above into the revised version.

---

> ### Author Response · Authors · 2024-08-13
> **Thank you, Reviewer LBha**
>
> Dear Reviewer LBha,
>
> We are glad that our rebuttal has successfully addressed your concerns. We will include the clarifications into the revised manuscript. Thank you again for your valuable comments.
>
> Best regards,
>
> Authors of Submission 2768

---

### Official Review · Reviewer_H1bu · 2024-07-12

**Soundness:** 3
**Presentation:** 3
**Contribution:** 2
**Rating:** 6
**Confidence:** 4

**Summary:**

This manuscript presents a new approach to detect fake videos generated with diffusion models. This approach is based on multimodality analysis and reports promising results on a database introduced by the author(s). These results are based on both frame and video levels. This work is well written and organized.

**Strengths:**

This poropsed approach sounds like a good novelty and shows its effectiveness with a good ablation study. As the type of generators increases, the methods can protect the users from the generated videos with fraudulent aspects in the real world. The users can verify the videos with the help of the methods.

**Weaknesses:**

A clear justification for the complex approach is not provided, and readers expect more details.

I think that the method is limited in a global application to the recognition of videos generated with diffusion models and that we should use other methods for other types.

**Questions:**

How can you add the method to a general method to detect all or most forgery video types?

What do you think will happen if you have created a video using other methods such as GAN methods?

**Limitations:**

These limitations are addressed by the author(s).

---

> ### Author Rebuttal · Authors · 2024-08-05
>
> We express our sincere gratitude to the reviewer for the insightful and valuable feedback.  We appreciate all constructive comments on the novelty, effectiveness, and clarity of the paper. We provide additional experiments and answer the weaknesses and questions mentioned in the review as follows.
>
> > **Q1**: A clear justification on MM-Det
>
> In our paper, We propose Multi-Modal Detection (MM-Det) for forgery detection. Briefly, the method integrates Multi-Modal Forgery Representations (MMFR) in the LMM branch with highly effective spatiotemporal features in the ST branch. In addition, A dynamic fusion strategy is applied to aggregate these features and adjust feature weights. Our proposed In-and-Across Frame Attention (IAFA) benefits our proposed method in effectively capturing forgery features. Besides, our method maintains a strong generalization ability on unseen forgery types with the help of large multimodal models.
>
> We provide detailed analyses and experiments for the functionality and effectiveness of our proposed MMFR and IAFA in the overall response. Specially, we provide
>
> 1) the functionality of IAFA compared with other baselines in Q1,
>
> 2) the generalization of our method on GAN and latest diffusion methods in Q2,
>
> 3) the performance of our method on different video durations and resolutions in Q2,
>
> 4) the ablation study on different LLMs for our method in Q3.
>
> > **Q2**: Generalization ability
>
> We provide additional performance results on GAN and latest diffusion methods in Q2 of the overall response to prove the generalization ability of our method on other types of forgery methods.
>
> > **Q3**: Improvements on more forgery content
>
> Additional performance results on GAN and latest diffusion methods in Q2 of the overall response prove the effectiveness of our method, showing that our finetuned large multimodal model specializes in forgery detection and demonstrates generalization ability on most forgery types. For adaptation to more forgery content,  enlarging the number and types of datasets and fine-tuning large multimodal models contributes to more promising performance.

---

> > ### Comment · Reviewer_H1bu · 2024-08-13
> >
> > Thank you for your response. I keep the rating.

---

> ### Author Response · Authors · 2024-08-13
> **Thank you, Reviewer H1bu**
>
> Dear Reviewer H1bu,
>
> Thank you for your recognition of our work. We greatly appreciate your valuable comments.
>
> Best regards,
>
> Authors of Submission 2768

---

### Official Review · Reviewer_f8wk · 2024-07-13

**Soundness:** 3
**Presentation:** 3
**Contribution:** 3
**Rating:** 6
**Confidence:** 5

**Summary:**

This paper proposes a method for detecting diffusion-generated fake videos using a multi-modal approach. Key contributions include:
A Multi-Modal Forgery Representation leveraging vision and language capabilities of large multimodal models
An In-and-Across frame attention to capture spatial-temporal forgery traces
A fusion strategy to combine multi-modal features
A new dataset of diffusion-generated videos for benchmarking
The method outperforms existing approaches on cross-dataset evaluation.

**Strengths:**

Creation of a new dataset to address lack of benchmarks in this area
Use of LMMs for video forgery detection, leveraging their visual and language understanding Combination of frame-level and video-level features
Evaluation on multiple diffusion video generation methods

**Weaknesses:**

The use of LMMs and complex Transformer architectures might impose high computational demands, which could limit practical deployment in resource-constrained environments.
Discussion on Failure Cases: The paper could benefit from a detailed discussion on scenarios where the proposed method fails or underperforms, which could guide future improvements and research.
Limited discussion of computational requirements and inference speed
Limited exploration of how the method generalizes to non-diffusion generated videos
it is unclear how the system performs under different operational conditions or with lower-quality videos.

**Questions:**

How does the computational cost and inference speed compare to existing methods?
How well does the method generalize to detecting non-diffusion generated fake videos?
Could the authors discuss the transferability of the proposed MMFR to other forms of synthetic media detection, such as audio or text?

**Limitations:**

Potential biases in the dataset
Scalability to longer videos or different resolutions
Adaptability to rapidly evolving diffusion models
Potential for overfitting to current generation artifacts

---

> ### Author Rebuttal · Authors · 2024-08-04
>
> We thank the reviewer for the valuable and insightful feedback. We appreciate all constructive comments on the novelty, soundness, and clarity of the paper. Here, we demonstrate additional experiments and answer the weaknesses and questions mentioned in the review.
>
> > **Q1**: Computational analysis
>
> As for computational requirements, our method is implemented on a single 4090 GPU with 24G memory for both training and inference. We provide further computational analysis and inference speed for our methods and other baselines in Table F.
>
> **Table F**: Comparison of computational cost and inference speed
> |Model|GFLOPs|Params|FPS|
> |--|--|--|--|
> |ViViT[1]|$84.1$|$26.4M$|$1201$|
> |TALL[2]|$30.4$|$86.6M$|$1445$|
> |UniversalFD[3]|$1556.4$|$304.5M$|$93$|
> |MM-Det(Ours)|$9345.2$|$6919M$|$40$|
>
> Compared with other baselines, our method reaches 6x GFLOPs and 23x Params of UniversalFD[3], which also uses a CLIP encoder for detection. As for inference speed, we conduct video-level inference at 40 fps. Regarding the computational cost and efficiency, we argue that the bottleneck of our method lies in the integration of the large multimodal model, which makes up 94% FLOPs in inference and 98% Params in total. With the improvement of LMM techniques and the appearance of more computational-friendly LMMs, such limitations will be solved. In addition, LMM inference on cloud service is available in practical deployment. Therefore, we hold the opinion that the computational cost of our method will be relieved to a large extent in the future. Further discussion is beyond the scope of our paper.
>
> > **Q2**: Generalization ability
>
> We report the performance of our method on multiple GAN-based methods in Q2 of the overall response, where we provide extensive experiments and results on
>
> 1) Generalization ability on non-diffusion videos,
>
> 2) Generalization ability on evolving diffusion videos,
>
> 3) Scalability to multiple durations and resolutions.
>
> With all promising performance, we prove that our method is effective in most cases.
>
> > **Q3**: Robustness analysis
>
> We provide additional experiments for robustness analysis in Table G.
>
> **Table G**: Performance of MM-Det on multiple post-processing in AUC
> |N/A|Blur $\\sigma=3$|JPEG $Q=50$|Resize $0.7$|Rotate $90$|Mixed|
> |:--:|:--:|:--:|:--:|:--:|:--:|
> |$95.5$|$89.2$|$93.2$|$91.7$|$92.1$|$91.9$|
>
> Our method remains effective under all such post-processing conditions.
>
> > **Q4**: Transferability to other modalities
>
> Our proposed MMFR takes advantage of feature spaces from multimodal encoders and text decoders in LMMs to form an effective representation for forgery detection tasks, which is a transferable method to other media. As long as the LMM is adapted to downstream tasks in audio or text, features from the encoder and decoder stand for a strong representation of unseen synthetic types. Therefore, our method is generalizable to other modalities.
>
> > **Q5**: Discussion on failure cases
>
> While our paper makes significant strides in addressing diffusion video detection, our exploration does not extensively cover partially manipulated content. For diffusion manipulations that only happen in small areas, our method may fail to capture informative features. A possible reason is that small forgery traces disappear after multiple downsampling operations in the deep network of LMMs. Future research endeavors may benefit from investigating the limitation to tackle the challenge of minor forgery detection.
>
> **References**
>
> [1] Arnab A, Dehghani M, Heigold G, et al. Vivit: A video vision transformer
>
> [2] Xu Y, Liang J, Jia G, et al. Tall: Thumbnail layout for deepfake video detection
>
> [3] Ojha U, Li Y, Lee Y J. Towards universal fake image detectors that generalize across generative models

---

### Official Review · Reviewer_X8oW · 2024-07-13

**Soundness:** 3
**Presentation:** 4
**Contribution:** 4
**Rating:** 6
**Confidence:** 4

**Summary:**

This paper proposed a video-level detection algorithm, named Multi-Modal Detection (MM-Det) for video forensics. MM-Det consists of Multi-Modal Forgery Representation (MMFR) that discriminates fake videos from real ones, In-and-Across Frame Attention (IAFA) that balances frame-level forgery traces with information flow across frames, and Dynamic Fusion Strategy for augmentation in features of high correlation and suppression for helpless ones integrates perception and reasoning abilities for video forensic work. In addition, this paper establishes a high-quality dataset including videos generated from various diffusion-based algorithms. Evaluation of several benchmarks confirms the effectiveness of MM-Det on general content from unseen diffusion models.

**Strengths:**

1.	The proposed multimodal representation fusion, In-and-Across Frame Attention, and Dynamic Fusion Strategy appear valid and reasonable.
2.	A new data set is proposed which is valuable for video forensics research.

**Weaknesses:**

1.	As for LLM, it isn't easy to see which module represents LLM from the paper and Figure 3. If LLM is decoupled and divided into several modules, the article should give explanations.
2.	The author declares that IAFA conducts global and local attention alternately. However, the paper lacks adequate analysis and experiments to illustrate the functionality of IAFA.
3.	Besides, LLM plays an important role in MM-Det. I think conducting an ablation study on LLM is necessary to see if different LLMs influence detection performance.

**Questions:**

1.	Which part in Figure 3 denotes LLM?
2.	Analysis and experiment to illustrate the functionality of IAFA.
3.	Ablation study on LLM.

**Limitations:**

The author has fully discussed the limitations of this work and pointed out the direction of future work.

---

> ### Author Rebuttal · Authors · 2024-08-04
>
> We extend our gratitude to the reviewer for the insightful feedback. We appreciate the constructive positive comments on the novelty, soundness, and valuable contribution of our paper. In response, we present additional experiments and address the weaknesses and questions highlighted in the review.
> > **Q1** As for LLM, it isn't easy to see which module represents LLM from the paper and Figure 3.
>
> We provide a more concrete explanation here. In our MM-Det framework, we apply a Large Multimodal Model to capture multimodal forgery representation, whose modules and outputs are denoted in dark blue in Figure 3 of our paper. The composition of the LMM branch is a typical structure of a multimodal large language model, including a tokenizer and an embedding layer for text input, a visual encoder for image input, and a transformer decoder layer as the language backbone.
>
> > **Q2** If LLM is decoupled and divided into several modules, the article should give explanations.
>
> Our proposed method obtains multimodal forgery representations (MMFR) by taking advantage of pretrained visual encoder and language decoder backbone from a LMM. Specifically, given an LMM composed of a vision encoder denoted as $\\mathcal{E}_v$, a text encoder denoted as $\\mathcal{E}_t$, and a language decoder backbone denoted as $\\mathcal{D}_t$. The input is a video sequence $\\{\\mathbf{x}\\}^N$ of $N$ frames and an instruction prompt $\\mathbf{p}$ from our forgery templates.
> We capture informative multimodal features $\\mathbf{F}_m$ by conducting text generation and extracting hidden states from the last layer of $\\mathcal{E}_v$ and $\\mathcal{D}_t$, which are feature representations in both visual and textual embedding space. This process can be expressed as:
> \begin{equation}\\mathbf{F}_v =\\mathcal{E}_v(\\mathbf{x})\\end{equation}
> \\begin{equation}\\mathbf{F}_t =\\mathcal{D}_t(\\mathbf{F}_v, \\mathcal{E}_t(\\mathbf{p}))\\end{equation}
> \\begin{equation}\\mathbf{F}_m=\\{\\mathbf{F}_v, \\mathbf{F}_t\\}\\end{equation}
> where $\\mathbf{F}_v$ and $\\mathbf{F}_t$ denote features in the visual and textual embedding space, respectively.
>
> > **Q3** Illustration of the functionality of IAFA
>
> We provide further analysis and demonstrate additional experiments in Q1 of the overall response.
>
> > **Q4** An ablation study on LLMs
>
> We provide an extensive ablation study on the influence of LLMs in Q3 of the overall response.

---

> > ### Comment · Reviewer_X8oW · 2024-08-13
> >
> > The additional experiments solved my concerns, so I have increased my score to 6 (weak accept).

---

> ### Author Response · Authors · 2024-08-13
> **Thank you, Reviewer X8oW**
>
> Dear Reviewer X8oW,
>
> We are glad that our rebuttal has addressed your concerns. Thanks again for your valuable comments.
>
> Best regards,
>
> Authors of Submission 2768

---

### Author Rebuttal · Authors · 2024-08-05

We appreciate all reviewers for their valuable comments and suggestions. We are delighted to see (a) all reviewers give positive feedback, (b) all reviewers recognize our proposed MMFR's novelty, (c) our MMDet achieves promising and generalizable performance on diffusion forgery detection (f8wk, H1bu, LBha, Niir), (d) a comprehensive dataset is proposed for future work (X8oW, f8wk, LBha, Niir) and (e) the insightful ablation study has been appreciated (H1bu). Moreover, we receive concerns about (a) illustration of the functionality of IAFA (X8oW, LBha, Niir), (b) MM-Det's generalization ability to more forgeries (f8wk, H1bu, LBha, Niir), and (c) additional justification on our MM-Det  (X8oW, H1bu, LBha, Niir). Therefore, we provide precise answers to all questions, which mainly include 1) additional analysis and experiments on IAFA, (2) experiments on the generalization ability of MM-Det, and (3) an ablation study on LLMs.
Additionally, please check our PDF attachment for more figures.

> **Q1**. Analysis and experiments of In-and-Across-Frame Attention

Spatio-temporal networks[1-3] have been discussed in the field of video-level tasks, such as video understanding and retrieval. These works utilize spatial and temporal attention to capture video-level information and represent the global attributes of videos. However, It is worth noticing that AI-generated videos often contain inconsistent frames, giving rise to forgery artifacts that happen on small periods or even single frames. This property makes it difficult for conventional video methods to capture local artifacts. To address this problem, we propose our effective attention mechanism IAFA, which specializes in local information aggregation.

The advantage of our IAFA over other baselines lies in that IAFA preserves local information at the frame level when conducting spatiotemporal attention, thus being effective in learning forgery features. Specifically, an additional temporal token is introduced in each frame for aggregation of frame-level forgery features. During forward propagation, our designed IAFA conducts in-and-across-frame attention to model both local and global forgeries at each frame consecutively.

To prove the effectiveness of IAFA, we compare the detection performance of IAFA with other spatiotemporal baselines in Table A. Our IAFA is based on a Hybrid ViT[4], and we select TS2-Net[1], ViViT[2], TALL[3], and Hybrid ViT[4] for comparison. Each model is trained on the same training dataset SVD from our proposed DVF, and evaluated on the rest diffusion datasets.

**Table A**: Comparison between IAFA and spatiotemporal baselines in AUC
|Model|Videocrafter|Zeroscope|Sora|Pika|Opensora|Stable Diffusion|Stable Video|Average|
|:--|:--:|:--:|:--:|:--:|:--:|:--:|:--:|:--:|
|Hybrid ViT|$72.3$|$70.3$|$67.3$|$88.4$|$53.6$|$86.0$|$74.1$|$73.1$|
|ViViT(CVPR2021)|$89.2$|$88.0$|$81.6$|$92.7$|$85.2$|$88.1$|$92.1$|$88.1$|
|TS2-Net(ECCV2022)|$60.7$|$72.0$|$81.0$|$80.2$|$74.3$|$60.2$|$80.2$|$72.7$|
|TALL(CVPR2023) |$76.5$|$61.8$|$62.3$|$79.9$|$69.8$|$85.9$|$64.8$|$71.6$|
|Hybrid ViT w/IAFA(Ours)|$94.4$|$94.2$|$82.0$|$95.4$|$82.0$|$92.8$|$93.9$|$90.6$|

Overall, our proposed IAFA achieves the best performance in the evaluation of unseen forgeries.

To further validate the functionality of IAFA on the preservation of local information, we provide the comparison of attention heat maps between IAFA and ViViT[2] in Figure A of our PDF material.

> **Q2**: Generalization ability

We provide an extensive performance of MM-Det on more generated videos in Table B. We choose 4 GAN-based methods[5-8] and one diffusion tool Kling (released on July 8th) for comparison. It is worthwhile noting that Kling was released after our submission.

**Table B**: Performance on GAN and diffusion videos in AUC
|StyleGAN-V[5]|StyleSV[6]|StyleSV-MTM[7]|TATS[8]|Kling| Average|
|:--:|:--:|:--:|:--:|:--:|:--:|
|$97.2$|$95.6$|$99.8$|$99.9$|$99.8$|$98.5$|

Our method gains competitive results, proving the generalization ability on unseen forgery types.

In addition, we report the performance of our method regarding video lengths and resolutions in Tables C and D.

**Table C**: Scalability on resolution in AUC
|1920x1080|1280x720|1024x576|512x512|
|:--:|:--:|:--:|:--:|
|$97.0$|$96.8$|$94.4$|$97.2$|

**Table D**: Scalability on duration in AUC
|[0, 2)s|[2, 10)s|[10, 20)s|>20s|
|:--:|:--:|:--:|:--:|
|$94.1$|$96.6$|$97.5$|$96.8$|

Our method is generalizable in multiple durations and resolutions.

> **Q3**: Ablation study on LLMs

We conduct an extensive ablation study on the influence of LLMs in Table E. We choose Vicuna-7b, Vicuna-13b and Mistral-7b as the LLM backbone.

**Table E**: Ablation study on LLMs in AUC
|LLM|Videocrafter|Zeroscope|Sora|Pika|Opensora|Stable Diffusion|Stable Video|Average|
|:--|:--:|:--:|:--:|:--:|:--:|:--:|:--:|:--:|
|N/A|$94.4$|$94.2$|$82.0$|$95.4$|$82.0$|$92.8$|$93.9$|$90.6$|
|Vicuna-7b|$99.2$|$98.4$|$97.4$|$95.5$|$99.4$|$98.0$|$98.4$|$98.0$|
|Vicuna-13b|$98.4$|$98.9$|$97.4$|$95.4$|$99.5$|$97.6$|$98.8$|$98.0$|
|Mistral-7b|$98.1$|$98.5$|$95.6$|$96.6$|$99.4$|$96.3$|$98.9$|$97.6$|

Overall, our proposed method is effective in different LLMs.

**References**

[1] Liu Y, Xiong P, Xu L, et al. Ts2-net: Token shift and selection transformer for text-video retrieval

[2] Arnab A, Dehghani M, Heigold G, et al. Vivit: A video vision transformer

[3] Xu Y, Liang J, Jia G, et al. Tall: Thumbnail layout for deepfake video detection

[4] Dosovitskiy A, Beyer L, Kolesnikov A, et al. An image is worth 16x16 words: Transformers for image recognition at scale

[5] Skorokhodov I, Tulyakov S, Elhoseiny M. Stylegan-v: A continuous video generator with the price, image quality and perks of stylegan2

[6] Zhang Q, Yang C, Shen Y, et al. Towards smooth video composition

[7] Yang C, Zhang Q, Xu Y, et al. Learning modulated transformation in GANs

[8] Ge S, Hayes T, Yang H, et al. Long video generation with time-agnostic vqgan and time-sensitive transformer

---

### Decision · Program_Chairs · 2024-09-25

**Decision:**

Accept (poster)

**Comment:**

This manuscript studied AI-Generated video detection based on LLM and learning multi-model forgery representation. A new dataset is also presented to address the lack of benchmarks in AIGC detection.
The four reviewers generally appreciate the contributions of this paper and give positive comments.
As for the weaknesses posed in the review process, the reviewers also are satisfied with the responses from the authors.
Hence, the AC suggests to accept this manuscript.